# Wiedemann–Steiner Syndrome: Case Report and Review of Literature

**DOI:** 10.3390/children9101545

**Published:** 2022-10-12

**Authors:** Huan Yu, Guijiao Zhang, Shengxu Yu, Wei Wu

**Affiliations:** Department of Pediatrics, Tongji Hospital, Tongji Medical College, Huazhong University of Science and Technology, Wuhan 430030, China

**Keywords:** KMT2A, intellectual disability, Wiedemann–Steiner syndrome

## Abstract

Wiedemann–Steiner syndrome (WDSTS) is an autosomal dominant disorder with a broad and variable phenotypic spectrum characterized by intellectual disability, prenatal and postnatal growth retardation, hypertrichosis, characteristic facial features, behavioral problems, and congenital anomalies involving different systems. Here, we report a five-year-old boy who was diagnosed with WDSTS based on the results of Trio-based whole-exome sequencing and an assessment of his clinical features. He had intellectual disability, short stature, hirsutism, and atypical facial features, including a low hairline, down-slanting palpebral fissures, hypertelorism, long eyelashes, broad and arching eyebrows, synophrys, a bulbous nose, a broad nasal tip, and dental/oral anomalies. However, not all individuals with WDSTS exhibit the classic phenotype, so the spectrum of the disorder can vary widely from relatively atypical facial features to multiple systemic symptoms. Here, we summarize the clinical and molecular spectrum, diagnosis and differential diagnosis, long-term management, and care planning of WDSTS to improve the awareness of both pediatricians and clinical geneticists and to promote the diagnosis and treatment of the disease.

## 1. Introduction

Wiedemann–Steiner syndrome (WDSTS, OMIM #605130) is a rare autosomal dominant disorder first described by Wiedemann et al. (1989) [1] and first defined as a syndrome by Steiner and Marques [2] (2000). Utilizing whole-exome sequencing (WES) in 2012, Jones et al. [3] identified that de novo mutations in the *KMT2A* gene were responsible for WDSTS in five of six studied individuals. The clinical phenotypes of WDSTS are complex, including developmental delay, intellectual disability (ID), hirsutism, and special facial features (long eyelashes, broad and arching eyebrows, a bulbous nose, and down-slanting and vertically narrow palpebral fissures). Less frequently accompanying clinical features include behavioral problems, congenital cardiac disease, gastrointestinal abnormalities, eye malformations, genitourinary defects, and immune and endocrine problems.

The *KMT2A* gene, a methyl-transferase for H3K4, plays an essential role in regulating gene expression during early development and hematopoiesis. KMT2A expression is ubiquitous throughout the body, and its highest expression occurs in the ovary, lymph node, brain, and skin tissue [4]. The encoded protein mediates chromatin modifications associated with epigenetic transcriptional activation, including the expression of multiple *Hox* and *Wnt* genes.

Currently, thanks to the development of molecular genetic testing, increasingly more patients are reported. This resulted in the WDSTS prevalence previously reported as 1 in 100,000 now being estimated at 1 in 25,000 to 40,000 [3,5]. However, there is still a lack of an international consensus, with increasing numbers of confirmed cases. In this study, we describe a five-year-old boy bearing a novel de novo *KMT2A* nonsense mutation (NM 001197104; c.7009C>T; p.Q2337X) as determined by trio-based WES (trio-WES). Additionally, to enhance the awareness of clinicians and promote the diagnosis and treatment of the disease, we review previously reported WDSTS cases.

## 2. Case Report

### 2.1. Clinical Description

We report the following case of a 5-year-old Chinese male who was born at term after an uneventful pregnancy. His birth weight was 2.95 kg, height was 50 cm, and data on his head circumference was not available. His parents were nonconsanguineous, and both were without significant clinical phenotypes. He sat independently at 8 months and crawled at 9 months. At the age of 18 months, he could walk independently. Despite his growth parameters being normal at birth, his postnatal growth and development lagged far behind his same-age peers. His mother complained that his weight and height started to drop at 24 months. At the age of 5, his weight, height, and BMI were 12 kg (<3rd percentile; −2.89 SD), 98.8 cm (<3rd percentile; −2.78 SD), and 12.3 kg/m^2^ (<3rd percentile; −2.95 SD), respectively. When he started kindergarten, his teacher noticed that the boy had an expressive language impairment and learning disability but relatively normal receptive language. The boy subsequently underwent a physical examination conducted by a specialist. His dysmorphic facial features (Figure 1a) and musculoskeletal phenotypes (Figure 1a–c) included a low hairline, down-slanting palpebral fissures, long eyelashes, thick and arching eyebrows, synophrys, a bulbous nose, a broad nasal tip, rib eversion flagging, and flat-footedness. His hair and hypertrichosis features (Figure 1b,c) were as follows: thick hair and hypertrichosis on the upper arms, legs, upper back, and lower back. His modified Ferriman–Gallwey (mFG) score (as a simpler predictor of total body hirsutism) was 16 [6]. Due to adamantly refusing to cooperate in the intelligence tests, the boy was unable to complete the administered IQ test. In combining the patient’s symptom complaints and clinical judgment, the patient was considered to have an intellectual disability (ID). His parents did not note any behavioral problems or seizures during his upbringing.

Laboratory examinations showed normal routine blood indexes, urine indexes, liver function, and kidney function. He also returned normal values for endocrine markers, including cortisol, insulin-like growth factor (76.10 ng/mL), insulin-like growth factor-binding protein-3 (1490 ng/mL), thyroid function, and reproductive hormones. To evaluate the patient’s growth hormone levels, he received growth hormone stimulation tests, which showed a growth hormone (GH) peak of 13.80 ng/mL. His electrocardiogram, echocardiography, abdominal ultrasonography, whole spine radiographs, and brain MRI were normal. However, the ultrasound of his testes showed left cryptorchidism with a normal right testis. Because of his ID, growth failure, hypertrichosis universalis, and facial features, the boy subsequently underwent a systematic and comprehensive genetic evaluation in the pediatric department of endocrinology, genetics, and metabolism.

### 2.2. Method

Genomic DNA was extracted from the peripheral venous blood of the patient and his parents. High-throughput sequencing was performed on the Illumina HiSeq X Ten system using the GenCap^®^ Whole Exon Gene Capture Probe V4.0 by Beijing MyGenostics Medical Laboratory (Beijing, China). The positional information of nucleotide changes refers to the GRCh37/hg19 genome version, and the REVEL (rare exome variant ensemble learner) software was used for protein function prediction, and pathogenicity as iws classified according to the variant interpretation guidelines issued by the American College of Medical Genetics and Genomics (ACMG) [6].

### 2.3. Result

The results of Tris-WES showed that the patient had a heterozygous nonsense mutation in the KMT2A gene c.7009C>T, and neither parent had the mutation, which was a de novo mutation. This variant has not been reported in these databases, including dbSNP (https://www.ncbi.nlm.nih.gov/snp/, accessed on 1 May 2022), ClinVar (https://www.ncbi.nlm.nih.gov/clinvar/, accessed on 1 May 2022), gnomAD (http://gnomad.broadinstitute.org/, accessed on 1 May 2022), ExAC (http://exac.broadinstitute.org/, accessed on 1 May 2022), 1000G (http://browser.1000genomes.org/, accessed on 1 May 2022), and HGMD (http://www.hgmd.cf.ac.uk/, accessed on 1 May 2022), suggesting that this is a novel mutation. This mutation resulted in a stop codon at position 2337 of glutamine (Gln). This variant was confirmed by Sanger sequencing (Figure 2a,b). According to the ACMG guidelines, this variant is preliminarily identified as a pathogenic variant (Pathogenic), and the evidence is PVS1, PS2, and PM2 criteria [6].

## 3. Review of the Literature

To obtain the relevant literature published before May 2022, several electronic databases were searched, including the Web of Science, PubMed, China National Knowledge Infrastructure (CNKI), VIP, and WanFang databases. The keywords used in the search were as follows: Wiedemann–Steiner syndrome, KMT2A, WSS, WSDTS, autism spectrum disorder, ASD, and ID/DD. Clinical data were collected from case reports and cohort studies. The inclusion/exclusion criteria were as follows: (a) all included patients had a genetic confirmation of the diagnosis; (b) patients without available clinical data were excluded; (c) patients with other genetic diseases were excluded; (d) patients diagnosed with acute myeloid and lymphoid leukemias were excluded. A total of 248 cases remained, including 132 males and 116 females. A total of 39 cases were reported in China and 209 were reported abroad. The age of diagnosis ranged from 4 months old to 46 years old, and approximately 85% of the patients were diagnosed before the age of 18. The analysis of these included 248 cases, including the single case reported within this study, are shown in Figure 3. For details, please see the Appendix A.

### 3.1. Facial Features

We reviewed 248 patients with WDSTS and found that the most common facial features were narrow palpebral fissures (119/161), down-slanting palpebral fissures (132/210), hypertelorism (163/214), long eyelashes (163/209), a wide nasal bridge (151/208), a depressed nasal bridge (34/47), a low hairline (40/49), a high palate (47/61), and dental/oral anomalies (46/65). However, the common facial features in the Chinese population are down-slanting palpebral fissures (19/25), hypertelorism (27/30), long eyelashes (26/27), ptosis (19/26), long philtrum (17/25), and dental/oral anomalies (13/24). Sheppard et al. conducted a multicenter, retrospective observational analysis of 104 people with WDSTS from five continents. This study was the first to identify race–facial feature associations and genotype–phenotype correlations in an ethnically diverse cohort, and found that the majority of participants had vertically narrow palpebral fissures (69.3%), hypertelorism (67.0%), and a wide nasal bridge (63.4%) with a broad or bulbous tip (63.6%) [7]. Baer et al. [8] stated that the classical facial features of this condition include vertically narrow down-slanting palpebral fissures, hypertelorism, thick eyebrows, long eyelashes, and a thin upper vermilion border. There was no apparent relationship in the race–facial feature associations, only some clinical differences in the probability of the occurrence of a given feature. In a recent publication, Sheppard et al. [7] cited that affected White people were more likely to have a thin upper vermillion border of the lip and a bifid uvula, whereas Black Indigenous People of Color (BIPOC) were more likely to feature an accentuated cupid’s bow of the lip. Li et al. [9] described the phenotypic characteristics of 16 patients with WDSTS and compared them with a large sample study of 33 cases in France. Ptosis, microcephaly, down-turned palpebral fissures, and long eyelashes were more frequent among the Chinese cohort while the frequency of a thin upper lip and thick eyebrows was higher in the French cohort. The single patient we reported on in this study presented with relatively atypical facial features. He just had mild down-slanting palpebral fissures without vertically narrow palpebral fissures. There was also an absence of any distinguishable hypertelorism, ptosis, a long philtrum, or dental/oral anomalies.

### 3.2. Hypertrichosis

The frequency of hirsutism in WDSTS patients is approximately 85.7% (72/84) and it is most commonly detectable via the screening of atypical facial features, including long eyelashes (166/209) and thick eyebrows (159/209). The rate of elbow hirsutism, being the most characteristic distinguishing feature of the condition, is 59.4% (120/202), which is less common than back hirsutism 73.2% (139/190). Although elbow hirsutism is specific, it is not present in all cases. When we compared the photographs of patients taken at different age classes, we found that the facial features and hirsutism gradually became more pronounced with increasing age, and such phenotypic indicators included down-slanting palpebral fissures, ptosis, synophrys, thick eyebrows, and thick hair [7,10,11,12,13,14,15,16].

### 3.3. Growth and Developmental Delays

Most patients exhibited different degrees of growth retardation and global developmental delay. Approximately 44.9% of the patients had prenatal growth retardation, 78.9% had postnatal growth retardation, and 92.8% had developmental delays. One study found that the median age for first sitting was 10 months, standing was 17 months, walking was 20 months, and first words was 18 months [7]. In severe cases, patients were unable to walk or speak.

### 3.4. Neurological Abnormalities

Almost all patients (97.2%, 211/217) had an ID, which is similar to findings reported by other cohort studies (100% [8]; 93% [9]; 97% [7]). In addition to ID, other neurodevelopmental disabilities were found to be prominent, including speech delay (78/88), behavioral disorders (74/183), hypotonia (126/181), and seizures (28/155). Due to the subjective preferences and knowledge gaps of their caregivers, the manifestations of autism spectrum disorder, aggressive behavior, hyperactivity, and so on are commonly ignored. The most common structural brain abnormality was found to be an abnormal corpus callosum or abnormal myelination by MRI [7,8,9]. However, some patients that had presented with neuropsychiatric symptoms did not have detectable structural brain abnormalities [7,8,9]. With respect to neurocognitive function, Rowena Ng et al. suggested that patients with WDSTS are more likely to show visuospatial defects and nonverbal reasoning through neurocognitive tests, whereas relative verbal skills remained unaffected [17]. Due to the fact that hippocampal function and cognitive function are highly correlated, these findings may provide clues for studying the neurogenesis of the hippocampal structure.

### 3.5. Musculoskeletal Features

Musculoskeletal problems are common, and these include scoliosis, craniovertebral junction anomalies, small and puffy hands and feet, brachydactyly, clinodactyly, tapering fingers with fetal finger pads, broad and short toes, rib anomalies, sacral dimples, and laxity of the distal joints. Some patients with severe scoliosis or hip dysplasia required surgery [18,19].

### 3.6. Senses and Dental/Oral Anomalies

Patients may also develop ophthalmologic abnormalities (myopia, astigmatism, hypermetropia, retinal atrophy), ear-nose-throat problems (obstructive sleep apnea, abnormal ears, hearing loss), and dental/oral anomalies (early dentition, delayed eruption of primary teeth, hypoplasia of the dental enamel, carious teeth, dental crowding, supernumerary teeth, and small teeth).

### 3.7. Gastrointestinal Disorders

Gastrointestinal disorders are a significant problem in WDSTS. In a study of 104 individuals with WDSTS, 63.8% had constipation, 66.3% had feeding difficulties, and 25.5% required tube feeding [7]. The causes of such feeding difficulties included food aversion, gastroesophageal reflux, and swallowing difficulties. Hiroyuki Iijima et al. reported that as gross motor skills develop, feeding disorders can improve due to symptoms of hypotonia resolving [20]. However, the relationship between improvements in hypotonia and the corresponding improvements in feeding disorders requires further investigation.

### 3.8. Cardiac Abnormalities

Cardiac abnormalities can occur in children with WDSTS, and these include patent ductus arteriosus, bicuspid aortic valve, dilated left ventricle, right aortic arc, aortic insufficiency, Laubry–Pezzi atrial septal defect, and cardiac arrhythmia, including the requirement for a pacemaker. The prevalence of cardiac abnormalities was reported to be 35.8% in a cohort of 104 individuals [7] and 36% in a cohort of 33 individuals [8], and these instances typically presented as structural cardiac anomalies (28.4%) [7].

### 3.9. Genitourinary Anomalies

Genitourinary anomalies in individuals with WDSTS include renal anomalies, uterine or testicular anomalies, and external genital anomalies. Moreover, a family genetic history of WDSTS indicates the speculation of normal fertility. Interestingly, only one of twelve families was found to have paternal transmission of WDSTS [7,8,9,21]. The reason for this phenomenon is currently unknown.

### 3.10. Endocrine Problems

With respect to the endocrine problems, 34.4% (*n*  =  32) of patients with WDSTS reported having premature adrenarche, and 18.8–50% reported growth hormone deficiency (GHD). Hypothyroidism, hypoglycemia, and menstrual issues have also been reported in the previous literature [7,8,9]. However, GHD can only partially explain short stature. A satisfactory height trajectory was observed in patients with growth hormone deficiency who received growth hormone therapy, and no adverse effects were noted. Patients with short stature and early puberty can be treated with gonadotrophin-releasing hormone (GnRH) and GH to inhibit the rapid sexual development and increase their final height.

### 3.11. Immunologic Issues

Immune dysfunction includes recurrent infections of the respiratory tract or urinary tract, poor vaccine responses, eosinophilia, and congenital immunodeficiency with low immunoglobulins levels [13,21,22].

## 4. Discussion

A total of two patients were diagnosed with a novel pathogenic mutation in the *KMT2A* gene in our hospital. Our first patient was a previously reported case of a 15-month-old girl [23]. She had growth delay, hypertrichosis, dysmorphic facies, hypotonia, and mild/moderate ID. She also had some characteristic phenotypes, such as swollen small and puffy hands, fat pads anterior to calcanei, and palmar/plantar grooves. As in the current report, our second patient was a 5-year-old boy with ID, facial dysmorphism, generalized hypertrichosis, and significant developmental delay. The boy also presented with relatively atypical facial features. The boy had no abnormal vital organ systems, except for left-sided cryptorchidism. Nevertheless, there remains the requirement for long-term follow-up assessments by an outpatient visit or by telephone because the phenotypic characteristics may develop in the boy over time. The detected *KMT2A* nonsense mutation (NM 001197104; c.7009C>T; p.Q2337X) is defined as a loss-of-function mutation and is thereby expected to result in a reduced transcript or protein level, ultimately leading to haploinsufficiency. Therefore, the haploinsufficiency of KMT2A might partially explain the clinical phenotype of this individual.

Regarding the pathogenesis of the disease, Léo Mietton et al. confirmed that the *KMT2A* gene mutation causes hirsutism by affecting endothelial nitric oxide levels and bone morphogenetic protein signaling pathways, and the resulting changes in the endothelial nitric oxide and Wnt signaling pathways may be related to cognition dysfunction [24]. KMT2A modulates the expression and H3K4 trimethylation of bone morphogenesis genes such as *Gdf6* (a bone morphogenesis gene that causes Klippel–Feil syndrome (KFS) [25]), *Pax1* (a KFS candidate gene [26]), and *Pax9* (a functionally relevant gene in vertebrate segmentation [27]). There are 322 reported *KMT2A* variants implicated in WDSTS (HGMD Professional Edition database), and these include missense, nonsense, frameshift, and splicing mutations. Most of these mutations result in a premature stop codon, which leads to either nonsense-mediated decay or the mutant protein degradation, thereby resulting in haploinsufficiency [3]. These studies jointly suggest that haploinsufficiency is central to the pathogenesis of WDSTS. Missense variants are more likely to be located in the CXXC domain, which is important for selectively binding to target genes containing unmethylated-CpG stretches [28]. Diana Ramirez-Montaño et al. [29] reviewed that more than 50% of pathogenic variants occur in exons 3 and 27, which represent the two longest exons of this gene; however, no mutation hotspots have been found. In our case, the mutation site detected in the boy was found to be located in exon 27.

About the relationship between the genotype and phenotype, this is not yet conclusive. Hypotonia was linked to loss-of-function (LoF) variations, and seizures were linked to non-LoF variants [7]. Li et al. suggested that missense variants occurred in the cysteine-rich CXXC zinc finger domain, which is associated with more severe neurophenotypes [9].

Due to the lack of existence of any clinical diagnostic criteria and clinical phenotypic heterogeneity, the diagnosis of WDSTS is established by molecular genetic testing, including gene-targeted testing (single-gene testing, multigene panel) and comprehensive genomic testing (exome sequencing, genome sequencing). For individuals suspected to have a genetic disease, clinicians can use bioinformatics tools to assist in the diagnosis. PhenIX (a computational method) can be used to evaluate and rank the possibility of an individual having particular diseases according to the Human Phenotype Ontology (https://hpo.jax.org/, accessed on 1 May 2022) of the patients [30]. Similarly, Face2Gene (an AI face recognition technology, https://www.face2gene.com/, accessed on 1 May 2022) can infer potential syndromes based on the facial photographs of individuals [31]. The rational use of bioinformatics tools not only narrows down the range of possible diseases but also saves both the time and costs involved in forming diagnoses.

Because of the wide variety of clinical phenotypes of WDSTS, the differential diagnosis based on phenotypic associations is difficult. Furthermore, we found that 43 disorders with hypertrichosis and ID are associated with genetic syndromes in the Online Mendelian Inheritance in Man (OMIM) database. To note, a few patients initially diagnosed with other chromatin diseases were finally diagnosed with WDSTS by genetic testing [32,33]. Owing to the broad clinical spectrum, overlapping features, and potentially common pathogenic molecular mechanisms, distinguishing WDSTS from typical epigenetic machinery disorders represents a significant challenge.

Coffin–Siris syndrome (CSS) is a multisystemic intellectual disability syndrome characterized by developmental or cognitive delays of varying degrees, distinctive facial features, musculoskeletal anomalies (aplasia or hypoplasia of the distal phalanx or nail of the fifth and additional digits), and organ system anomalies. CSS is caused by mutations in genes that encode BRG1(BRM)-associated factors (BAF, Brahma-associated factor) complex, including *ARID1B*, *ARID1A*, *SMARCB1*, *SMARCA4*, *SMARCE1*, *ARID2*, *DFP2*, *SMARCC2*, *SOX11*, *SOX4*, *SMARCD1*, and *BICRA* [34,35]. A rare developmental disorder known as Kabuki syndrome (KS) is characterized by postnatal growth restriction, characteristic facial features (such as long palpebral fissures in the lateral third of the lower eyelids, arched and broad eyebrows with the lateral third displaying notches or sparseness, large, prominent, or cupped ears, and short columella with a depressed nasal tip), skeletal anomalies (such as brachymesophalangy, brachydactyly V, spinal column abnormalities, and fifth digit clinodactyly), dermatoglyphic abnormalities (persistent fingertip pads), and mild-to-moderate intellectual disability [36]. KS is primarily (60%) caused by mutations in *KMT2D*, or less often in *KDM6A* [36,37]. Cornelia De Lange syndrome (CdLS) is an archetypical genetic syndrome characterized by ID, distinctive facial features, malformations of the upper limbs, and atypical growth, among numerous other signs and symptoms. CdLS is primarily (70%) caused by mutations in genes that encode subunits or regulators of the cohesin complex, including *NIPBL*, *SMC1A*, *SMC3*, *RAD21*, and *HDAC8*, and in fewer cases *BRD4* and *ANKRD11* [38,39]. Rubinstein–Taybi syndrome (RSTS, OMIM*180849) is a rare autosomal dominant genetic disorder, characterized by distinctive facial features (down-slanting palpebral fissures, low hanging columella), skeletal abnormalities (microcephaly, broad and often angulated thumbs and halluces), short stature, ID, and occasional congenital anomalies. RSTS is caused by pathogenic variants in the genes *CREBBP* and *EP300* [40].

When considering a diagnosis of WDSTS, it is recommended that other associated abnormalities are also further evaluated.

The clinical guidelines proposed by Sheppard et al. [7] and Baer et al. [8] suggest performing a CT scan or brain MRI of the vertebral block and medullar consequences, EEG to detect signs of seizures, and screening of non-LoF variants. Despite significant advances in the recognition and diagnosis of the disease, current treatments are mainly based on corresponding medical and surgical interventions. Cardiac functions are assessed by echocardiography and electrocardiogram. Endocrinological evaluations include measurements of growth hormone, thyroid hormone, and other indicators of metabolic bone disease. It has been reported that 18.8–50% of children with WDSTS are deficient in GH, so it is necessary to evaluate GH secretion using GH stimulation testing [41]. Additional screening for gastrointestinal and genitourinary anomalies with abdominal ultrasound should also be considered. Other screening methods may involve immunology, vertebral column analysis, ophthalmology, otolaryngology, and dentistry. Growth, feeding difficulties, and development also need to be monitored in the long term.

To better assist affected patients in managing their overall health, multidisciplinary and comprehensive intervention requires a multidisciplinary team that is tailored to the needs of each patient. Patients with global developmental delays might require special education and appropriate assessments at school. Some studies [9,41] have shown that rhGH can be used successfully and safely to improve growth and increase adult height in patients with short stature regardless of growth hormone deficiency in the absence of contraindications. GH treatment and management follow the consensus-based clinical practice guidelines of the Growth Hormone Research Society (GRS) [42]. Additionally, physical therapy, occupational therapy, speech therapy, and early intervention play a crucial role. When WDSTS affects multiple organ systems, standard treatments are often required by the relevant specialist.

## 5. Conclusions

In summary, we reported on a patient with a nonsense pathogenic variant that has not been previously reported in the *KMT2A* gene, thereby extending the spectrum of disease phenotypes associated with *KMT2A* mutations. We summarized the clinical and molecular spectrum, diagnosis and differential diagnosis, treatment, and management experiences of WDSTS to enhance the knowledge of this rare condition. Due to the large phenotypic spectrum of WDSTS patients, it remains challenging to draw more conclusions about the range of genotypes and phenotypes. Additionally, there is substantial phenotypic overlap between WDSTS and several other chromatinopathies (e.g., growth retardation, ID (mild to severe), facial features, and skeletal abnormalities). Furthermore, there are very few treatment therapies available for WDSTS patients. In conclusion, clinicians ought to provide biochemical and genetic tests for patients with developmental delay, ID, hirsutism, and special facial traits and actively evaluate the possibility of *KMT2A* mutation.

## Figures and Tables

**Figure 1 children-09-01545-f001:**
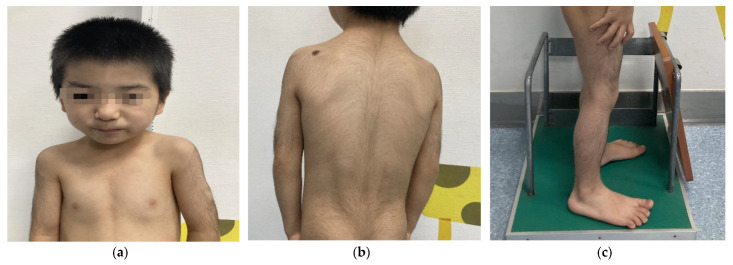
Physical characteristics of a 5-year-old boy with WDSTS. (**a**) Thick hair, rib eversion flagging, and special facial features, including low hairline, down-slanting palpebral fissures, long eyelashes, broad and arching brows, synophrys, bulbous nose, and a large nasal tip. (**b**) Hypertrichosis on the back and upper limbs. (**c**) Hypertrichosis on the lower limbs.

**Figure 2 children-09-01545-f002:**
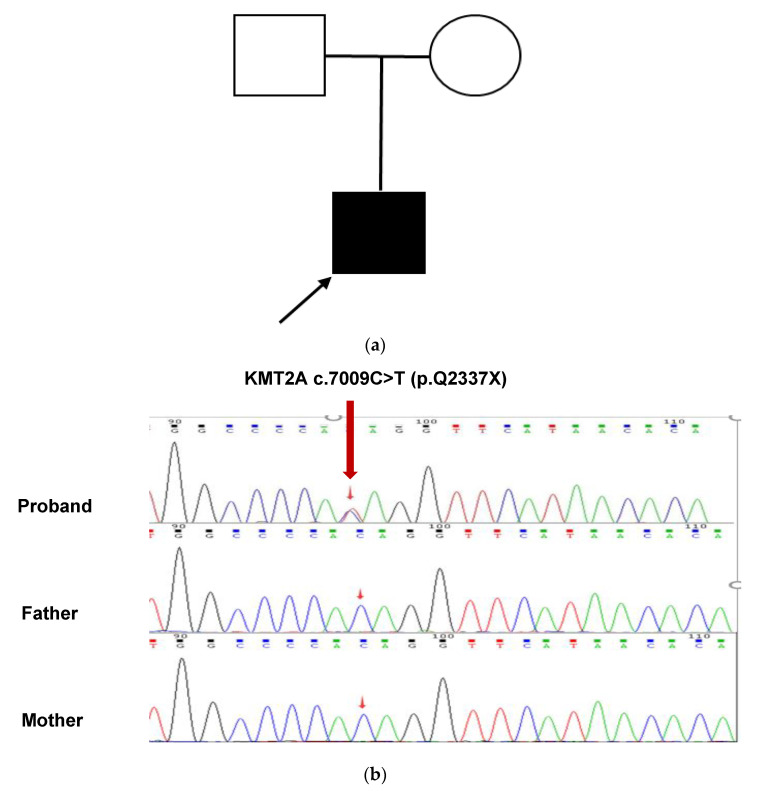
Analysis of the pathogenic genetic mutation in the patient (**a**) This patient’s family tree. The unaffected man and female are represented by a blank square and circle, respectively. The filled square represents the afflicted male while the arrow represents the proband. (**b**) Sanger sequencing confirmed a heterozygous KMT2A c.7009C>T (red arrow). At this nucleotide, both parents had the wild-type sequence.

**Figure 3 children-09-01545-f003:**
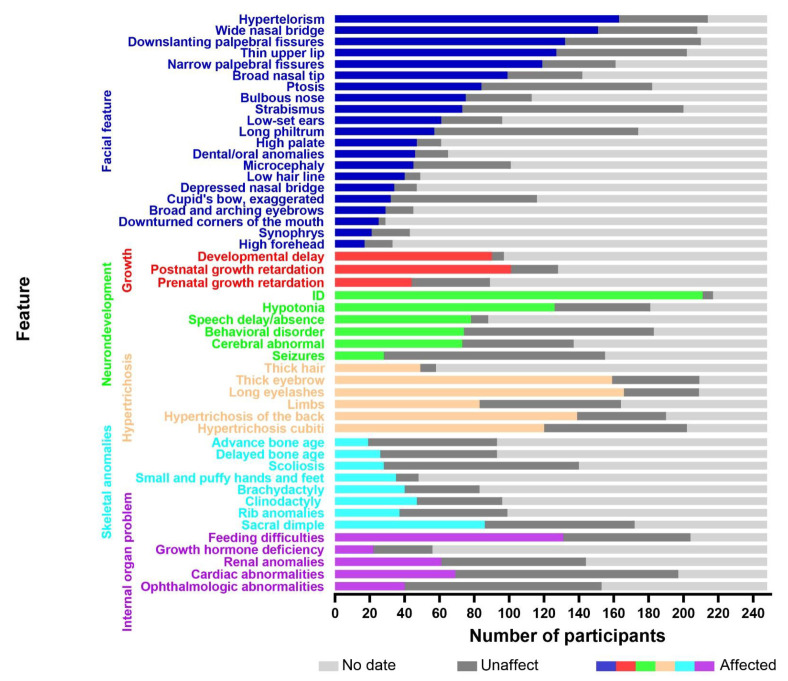
The *X*-axis represents the number of participants and the *Y*-axis represents the clinical features. The colored boxes indicate affected individuals, the dark gray boxes indicate unaffected individuals, and the light gray boxes denote data not available.

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
