# Peer review of "Wiedemann–Steiner Syndrome: Case Report and Review of Literature"

_children, 2022, doi:10.3390/children9101545_

Round 1

Reviewer 1 Report

Although authors describe another patient with Wiedemann Steiner Syndrome (WSS) due to a novel KMT2A mutation, this paper lacks both a deep clinical description and an incisive overview among previously reported other cases, being them only listed. Therefore, before considering this article for publication, an extensive review should be performed, concerning both the English language and the contents.

Major concerns:

  1. Introduction should be enriched.

  2. Case report: neurodevelopmental milestones should be detailed.

  3. Case report, page 2, line 55: which test had been administered in order to diagnose an ID? What authors mean with "Speech delay"? Did the child utter few words or was the language absent? Had autism spectrum disorder been considered in the diagnostic process?

  4. Considering that the authors' patient carried a loss-of-function mutation, they should specify if EEG and echocardiography/ECG, complete abdominal ultrasound, and ophthalmologic evaluation had been executed; otherwise, they should explain the reason of their decision.

  5. Had craniovertebral junction and spinal cord been studied by imaging techniques?

  6. Page 2, Figure 1, line 72: "rib eversion flagging" is not a facial feature.

  7. The Discussion section is more similar to a Results section, and it lacks imbrications among the different features reported, and considerations about gene function and its effects on phenotype. In particular, authors should point out any relevant difference between their patient's phenotype and what is already known about WSS, given that they underlined phenotypic spectrum variability and differences between the "classic" phenotype and others.

  8. Discussion, page 5-6: it is necessary to better relate WSS to other overlapping syndromes, such as chromatinopathies. In the current form, it appears instead as a pleonastic list of clinical features and genes without any relation, and out-of-focus compared to the argument. In addition, authors should analyze other possible diagnostic methods than WES to reach the genetic confirmation for WSS and other differential diagnoses.

  9. Discussion, page 6, line 204, DPF2 is the correct name of the gene.

  10. Discussion, page 6, line 218, ANKRD11 mutations provoke KBG syndrome.

  11. Among references, the following should be added and discussed:

-Grangeia et al. Wiedemann-Steiner syndrome in two patients from Portugal. Am J Med Genet A. 2020 Jan;182(1):25-28.

-Koenig R et al. Wiedemann-Steiner syndrome: three further cases. Am J Med Genet A. 2010 Sep;152A(9):2372-5.

-Fallah MS et al. Impaired Regulation of Histone Methylation and Acetylation Underlies Specific Neurodevelopmental Disorders. Front Genet. 2021 Jan 8;11:613098.

-Dunkerton S et al. A de novo Mutation in KMT2A (MLL) in monozygotic twins with Wiedemann-Steiner syndrome. Am J Med Genet A. 2015 Sep;167A(9):2182-7.

-Mendelsohn BA et al. Advanced bone age in a girl with Wiedemann-Steiner syndrome and an exonic deletion in KMT2A (MLL). Am J Med Genet A. 2014 Aug;164A(8):2079-83.

-Foroutan A et al. Clinical Utility of a Unique Genome-Wide DNA Methylation Signature for KMT2A-Related Syndrome. Int J Mol Sci. 2022 Feb 5;23(3):1815.

-Strom SP et al. De Novo variants in the KMT2A (MLL) gene causing atypical Wiedemann-Steiner syndrome in two unrelated individuals identified by clinical exome sequencing. BMC Med Genet. 2014 May 1;15:49.

Minor concerns:

  1. Case report, page 2, line 44: the term "unclear" is vague.

  2. Case report, page 2, line 49: BMI measure unit is missing.

  3. Discussion, page 5, line 171, the word "people" is repeated.

  4. Supplementary tables should be more clear. Please avoid Chinese characters and extend "Giangiobbe et al. 2020" as header to the whole line, if it is correct.

  5. The abbreviation WSS has never been written in extenso throughout the manuscript. Furthermore, authors use firstly WDSTS, then WSS, subsequently WDSTS. This inconsistency is misleading.

Author Response

It is with excitement that I resubmit to you a revised version of manuscript (children-1912335)for the Children. Thank you for giving me the opportunity to revise and resubmit this manuscript. In keeping with my last communication with you, I am resubmitting this revision before the agreed upon deadline. I appreciate the time and detail provided by each reviewer and by you and have incorporated the suggested changes into the manuscript to the best of my ability. The manuscript has certainly benefited from these insightful revision suggestions. I look forward to working with you and the reviewers to move this manuscript closer to publication in the Children.

Reviewer 2 Report

The authors had submitted this article as a review article regarding Wiedemann-Steiner Syndrome (WDSTS), aiming to summarize the clinical and molecular spectrum, diagnosis and differential diagnosis, long-term management and care planning of WDSTS to improve the awareness of paediatricians and clinical geneticists and to promote the diagnosis and treatment of this disease.

However, the article style is not that of a review article but of a case report. I strongly recommend the authors to change the style of this article, including a three-year-old boy with WDSTS as  a sample case, if they are keen to submit it as a review article.

The part of Review of literature and Discussion should be unified and reorganized with some  sub-headings.

The authors' conclusion seemed to be that clinicians ought to provide biochemical and genetic tests for patients with developmental delay, ID, hirsutism, and special facial traits as well as actively evaluate the possibility of KMT2A mutation. It would be better for the authors to focus on some issues to draw this conclusion. Because the descriptions in Discussion are too redundant.

Author Response

(The authors gave the same response as above.)

Reviewer 3 Report

The authors present an interesting case of Wiedemann-Steiner syndrome detected by trio exome sequencing and provide a thorough review of the literature, including previous reported cases, related disorders, and clinical management recommendations.

Specific comments:

- If room allows, consider incorporating the supplemental table into the main body of the paper, in addition to the existing figures or in place of figure 2 (which adds little visually to the case report).

- Please italicize all uses of the gene name (KMT2A)

- Lines 48-49: Z-scores would be helpful in order to judge how far below the 3rd percentile each of these values are.

- Lines 55-56: I would be interested in knowing more about the developmental history and degree of intellectual disability.

- Lines 113-141: Percentages to the hundredths place (e.g. 84.68%) seems excessive; I would consider rounding to the tenth (84.7%) or even integer (85%).

Round 2

Reviewer 1 Report

The paper has been significantly ameliorated, although some points should be further pointed out:

Page 2, Line 84, the abbreviation ID should be written after having specified its corresponding form in extenso (line 72). Consequently, this information can be omitted at line 203.

Page 3, Figure 1: was the boy 5 or 3 years old?

Figure 3: legend colors should be clarified and corrected to be better understandable.

Page 7, Line 248, please specify GHD in the extended version before.

Page 8, line 264, Authors state again that "The boy also presented with relatively atypical facial features", although the phenotype they described is quite overlapping with the classic. In the response to the reviewer they specified "The single patient we reported on in this study presented with relatively atypical facial features. He just had mild down-slanting palpebral fissures without vertically narrow palpebral fissures. There was also an absence of any distinguishable hypertelorism, ptosis, a long philtrum, or dental/oral anomalies". It should be better if they add this information in the text in order to explain why their patient's features are atypical.

Page 8, Line 273-277: please add considerations about your non-sense mutation distinguishing its effects from those of a missense mutation.

Page 10, please summarize the overlapping syndromes underlining similarities and differences with WDSTS.

Author Response

It is with excitement that I resubmit to you a revised version of manuscript children-1912335 for the Children. Thank you for giving me the opportunity to revise and resubmit this manuscript. I appreciate the time and detail provided by each reviewer and by you and have incorporated the suggested changes into the manuscript to the best of my ability. The manuscript has certainly benefited from these insightful revision suggestions. I look forward to working with you and the reviewers to move this manuscript closer to publication in the Children.

Reviewer 2 Report

My comments are as follows.

The authors reported in this manuscript a five-year-old boy who was diagnosed with Wiedemann-Steiner Syndrome (WDSTS) based on the results of Trio-based whole exome sequencing and an assessment of his clinical features and the review of literatures regarding WDSTS in 248 cases. Although I agree that the variety of information of this rare condition could be useful for readers of this journal, I have some concerns as follows.

1.    I consider that the sections of “3. Method and result” and “4. Review of the literature” describing DNA analyses are better to be included in 2. Case report.

2.   The contents of 4. Review of the literature are better to be separated with subheadings, such as Physical characteristics, Growth retardation and global developmental delay, Intellectual disability (ID), Musculoskeletal problems, Ophthalmologic abnormalities, Gastrointestinal disorders, Cardiac abnormalities, Genitourinary anomalies, Others, so on, which may be helpful for reader to understand them.

3.     Although the authors described “We summarize the clinical and molecular spectrum, diagnosis and differential diagnosis, treatment and management experiences of WDSTS to enhance the knowledge of this rare condition”, I was not able to find the appropriate description regarding treatment and management of WDSTS in this manuscript.

Author Response

(The authors gave the same response as above.)
